# Characterization and Modification of Light-Sensitive Phosphodiesterases from Choanoflagellates

**DOI:** 10.3390/biom12010088

**Published:** 2022-01-06

**Authors:** Yuehui Tian, Shang Yang, Georg Nagel, Shiqiang Gao

**Affiliations:** 1Environmental Microbiomics Research Center, School of Environmental Science and Engineering, Southern Marine Science and Engineering Guangdong Laboratory (Zhuhai), Sun Yat-sen University, Guangzhou 510006, China; tianyh8@mail.sysu.edu.cn; 2Department of Neurophysiology, Institute of Physiology, Biocenter, University of Wuerzburg, 97070 Wuerzburg, Germany; shang.yang@uni-wuerzburg.de

**Keywords:** choanoflagellates, optogenetics, rhodopsin phosphodiesterase (RhoPDE), cGMP

## Abstract

Enzyme rhodopsins, including cyclase opsins (Cyclops) and rhodopsin phosphodiesterases (RhoPDEs), were recently discovered in fungi, algae and protists. In contrast to the well-developed light-gated guanylyl/adenylyl cyclases as optogenetic tools, ideal light-regulated phosphodiesterases are still in demand. Here, we investigated and engineered the RhoPDEs from *Salpingoeca rosetta*, *Choanoeca flexa* and three other protists. All the RhoPDEs (fused with a cytosolic N-terminal YFP tag) can be expressed in *Xenopus* oocytes, except the *As*RhoPDE that lacks the retinal-binding lysine residue in the last (8th) transmembrane helix. An N296K mutation of YFP::*As*RhoPDE enabled its expression in oocytes, but this mutant still has no cGMP hydrolysis activity. Among the RhoPDEs tested, *Sr*RhoPDE, *Cf*RhoPDE1, 4 and *Mr*RhoPDE exhibited light-enhanced cGMP hydrolysis activity. Engineering *Sr*RhoPDE, we obtained two single point mutants, L623F and E657Q, in the C-terminal catalytic domain, which showed ~40 times decreased cGMP hydrolysis activity without affecting the light activation ratio. The molecular characterization and modification will aid in developing ideal light-regulated phosphodiesterase tools in the future.

## 1. Introduction

Rhodopsins are uniquely light-sensitive integral membrane proteins, sensing over a wide range of the visible spectrum, from UV to red light. The type I microbial rhodopsins were originally discovered to comprise seven transmembrane helices and function as ion channels or pumps, which directly convert light stimulation to a change in the electrochemical potential [1,2]. This enables the application of these light-sensitive proteins as optogenetic tools. Remarkably, Channelrhodopsin-1 (ChR1) and Channelrhodopsin-2 (ChR2) from the green alga *Chlamydomonas reinhardtii* were the first light-gated cation channels [1,3] discovered, enabling light-dependent depolarization of different cell types. Thus, ChR2 and its variants became powerful optogenetic tools in neuroscience [4,5,6,7]. 

We proposed that type I rhodopsins should be further divided into type Ia and type Ib, where the type Ia rhodopsins are composed of seven transmembrane helices (7-TM) with an extracellular N-terminus, while the type Ib rhodopsins comprise eight transmembrane (8-TM) helices with both termini on the cytosolic side [8]. Most characterized type Ib rhodopsins were recently discovered from fungi, green algae and protists, etc., and exhibited light-regulated enzyme activities, producing or degrading the second messenger cGMP and cAMP. For example, *Be*Cyclop (also named Rh-GC), identified in *Blastocladiella emersonii,* was characterized as a green light-activated guanylyl cyclase [9,10,11]. Later, the two-component cyclase opsins *Cr*2c-Cyclop1 and *Vc*2c-Cyclop1 were characterized as green light-inhibited guanylyl cyclases from *C. reinhardtii* and *Volvox carteri* [8]. Then, a chimeric switch-Cyclop1, activated by UV and inhibited by blue light, was generated by molecular engineering [12]. In protists, *Sr*RhoPDE (also named *Sr*Rh-PDE) was first identified and then proven as blue light-activated phosphodiesterase, degrading cGMP and cAMP [13,14,15]. The crystalized *Sr*RhoPDE structure confirmed a 8-TM topology, where the extra TM0 plays a key role in the photoactivity of RhoPDE [16]. However, the *Sr*RhoPDE shows considerable dark activity (turnover ~12 ± 2 s^−^^1^), which is enhanced to ~28 ± 5 s^−^^1^ by illumination [15]. This makes it difficult to be applied as an ideal optogenetic tool. 

Interestingly, more RhoPDEs were identified from different species of choanoflagellates. In *Choanoeca flexa* sp. nov., four homologous RhoPDEs (here named *Cf*RhoPDE1 = *Cf*Rh-PDE1, *Cf*RhoPDE2 = *Cf*Rh-PDE2, *Cf*RhoPDE3 = *Cf*Rh-PDE3, *Cf*RhoPDE4 = *Cf*Rh-PDE4) were discovered to play roles in regulating the curvature of colonies via a light-stimulated rhodopsin-cGMP pathway [17]. Moreover, from sequenced transcriptome databases, another four RhoPDE homologs were identified in *Microstomoeca roanoka* (named *Mr*RhoPDE = *Mr*Rh-PDE), *Acanthoeca spectabilis* (named *As*RhoPDE = *As*Rh-PDE) and *Choanoeca perplexa* (named *Cp*RhoPDE1 = *Cp*Rh-PDE1, *Cp*RhoPDE2 = *Cp*Rh-PDE2) [18]. *Cf*Rh-PDE1, *Cf*Rh-PDE4 and *Mr*Rh-PDE were characterized as light-regulated PDEs [19]. 

Here, we expressed the above eight RhoPDEs, fused with a YFP tag at the N-terminus, in *Xenopus* oocytes. After in vitro assay, we observed that only YFP::*Cf*RhoPDE1, YFP::*Cf*RhoPDE4 and YFP::*Mr*RhoPDE showed cGMP hydrolysis activity, where light/dark activity (L/D) ratios ranged from 1.3 to 3.5, depending on different initial substrate cGMP concentrations. By sequence analysis and engineering, we further identified sequences with different expression or activities. The *As*RhoPDE N296K mutant enables expression but still shows no cGMP hydrolysis activity. A short sequence in the catalytic domain of *Cf*Rh-PDE2 might be redundant, and its deletion recovered cGMP hydrolysis activity. Furthermore, we generated two *Sr*RhoPDE point mutations, L623F and E657Q, in the catalytic domain. Both showed ~30–40 times decreased cGMP hydrolysis activity with unchanged L/D ratio. These mutants with reduced dark activity may be useful in certain cells as optogenetic tools.

## 2. Materials and Methods

### 2.1. Gene Synthesis and Generation of Constructs

The sequence information of RhoPDEs were kindly provided by Dr. N. King of UC Berkeley. The DNA sequences of all the eight RhoPDEs (including *Cf*RhoPDE1-4, *Mr*RhoPDE, *As*RhoPDE, *Cp*RhoPDE1-2) were codon-optimized to *Mus musculus* and synthesized by GeneArt Strings DNA Fragments (Life Technologies, Thermo Fisher Scientific, Waltham, MA USA). The synthetic RhoPDE fragments were all designed with a 5′-XhoI restriction site and 3′-HindIII restriction site. The vector pGEMHE with 5′-BamHI-YFP (no stop codon) -XhoI-HindIII-3′ was used as the backbone. The YFP::RhoPDE constructs were generated by DNA ligation after digestion of vector and fragments with XhoI and HindIII restriction enzymes.

The mutations, including YFP::*Cf*RhoPDE2 (28 AA-deletion, 578-VRPKTFTSAAQKDYEIHPVGFHCAFVPQ-605 briefly named YFP::*Cf*RhoPDE2-del), YFP::*Cf*RhoPDE3 mutant (441-LRNVG-445 to 441-CHDCD-445), YFP::*As*RhoPDE N296K, YFP::*Cp*RhoPDE1 mutant (436-LRNVG-440 to 436-CHDCD-440), *Sr*RhoPDE L623F and E657Q mutants, were generated by QuikChange site-directed mutagenesis methods. We used the primers to replicate the plasmid DNA, and then mixed Dpn I in the PCR mixture to digest the parental methylated vector with a wild type RhoPDE fragment. Therefore, the whole vector with the mutated sites was generated. The list of primer sequences is shown in Table 1.

Sequences of all the constructs were confirmed by complete DNA sequencing. A NheI restriction enzyme was used to linearize all the plasmids before the in vitro transcription to generate cRNA by the AmpliCap-MaxT7 High Yield Message Maker Kit (Epicentre Biotechnologies).

### 2.2. Expression in Xenopus Oocytes

A certain amount of cRNA (10–30 ng, indicated in individual figure) was injected into *Xenopus* oocytes. Then, the oocytes were collected and incubated in the dark for 2–3 days with ND96 buffer (96 mM NaCl, 5 mM KCl, 1 mM CaCl_2_, 1 mM MgCl_2_, 5 mM HEPES, pH adjusted to 7.6) in an 18 °C incubator. Additional 1 µM all-trans retinal (ATR) was supplemented in ND96 buffer.

### 2.3. Xenopus Oocytes Membrane Extraction and In Vitro PDE Activity Assay

The YFP::RhoPDEs were heterologously expressed in the oocytes for 2–3 days. The membrane fraction of oocytes was extracted according to ref. [10]. Solution A buffer (including 100 mM NaCl, 75 mM Tris, 5 mM DTT, 5% glycerol, pH 7.3 adjusted with HCl) was used to extract and resuspend membranes. Each oocyte membrane was resuspended with 8 µL solution A. During in vitro reaction, we mixed the membrane suspension with reaction buffer in a ratio of 1:9 with the initial substrate cGMP concentration at 1–100 µM (indicated in figures). The reaction buffer contained 100 mM NaCl, 5 mM MgCl_2_, 75 mM Tris–HCl, 5 mM DTT, pH adjusted to 7.3, 1–100 µM cGMP concentrations as indicated. To stop in vitro reaction, aliquots of 10 µL in each time point were taken and immediately mixed with 190 µL stop solution. Under dark or light conditions, the decreasing concentrations of cGMP was measured by a Chemiluminescent Immunoassay Kit (Arbor Assays) at certain time points, so the light and dark activity and L/D activity ratio could be calculated. 

### 2.4. Fluorescence Emission Detection

The fluorescence emission values of oocytes expressing YFP-fused RhoPDE and mutants were detected by a Fluoroskan Ascent microplate fluorometer. The excitation and emission wavelengths were applied at 485 nm and 538 nm, respectively. The protein amount can be calibrated by a standard curve of quantified YFP proteins.

### 2.5. Illumination Conditions

Illuminations were mainly operated by a 505 nm (±10 nm) LED (ProLight Opto Technology, Taiwan, China) for testing activities of the eight RhoPDEs and their mutants. A 473 nm blue laser (Changchun New Industries Optoelectronics Tech, Changchun, China) was applied for testing the cGMP hydrolysis activity of the L623F and E657Q mutants of *Sr*RhoPDE. The light intensities were detected by a Plus 2 optical power and energy meter (LaserPoint s.r.l, Vimodrone, Italy).

### 2.6. Imaging of Xenopus Oocytes

Fluorescence images of oocytes were taken by a Leica DM6 M LIBS Microscope at 3 days after YFP::RhoPDEs expression in *Xenopus* oocytes. We conducted imaging on live cells.

### 2.7. Data Analysis

The individual measurement was repeated at least three times for the in vitro reaction. All data were analyzed by Origin 2018 and Microsoft Excel. The mean values and standard deviation were presented in figures.

### 2.8. Bioinformatics

The Clustal Omega 1.2.2 and Genedoc were applied to perform the sequence alignment and compare the conserved residues among certain protein fragments. The prediction of eight transmembrane helices in rhodopsin domains of RhoPDEs were performed by TMHMM [20,21] and JPred4 [22].

## 3. Results

### 3.1. Comparison of RhoPDEs

In a previous study, we characterized the *Sr*RhoPDE from Choanoflagellate *Salpingoeca rosetta* to be a blue light-stimulated phosphodiesterase that can degrade both cGMP and cAMP [15]. Here, we studied another eight RhoPDEs from four different protists, *Cf*RhoPDE1, *Cf*RhoPDE2, *Cf*RhoPDE3 and *Cf*RhoPDE4 from *C. flexa*, *Mr*RhoPDE from *M. roanoka*, *As*RhoPDE from *A. spectabilis*, *Cp*RhoPDE1 and *Cp*RhoPDE2 from *C. perplexa*. Alignment with *Sr*RhoPDE showed that *Cf*RhoPDE1 and *Mr*RhoPDE had a greater similarity to *Sr*RhoPDE at 81% and 91% (with identity at 63% and 79%), while the *As*RhoPDE had the lowest similarity at 49% (Table 2).

Besides the general homology, some key amino acids were also identified to be different for four RhoPDEs in comparison to *Sr*RhoPDE. For example, a short sequence of 28 amino acids seemed to be redundant in the catalytic domain of *Cf*RhoPDE2 (Table 2). The Mg^2+^-binding site was conserved in the PDE catalytic domains as an aspartate (D), but *Cf*RhoPDE3 and *Cp*RhoPDE1 exhibited a different amino acid, asparagine (N), in the corresponding positions. For the *As*RhoPDE, the key retinal-binding site, a lysine residue, was replaced by an asparagine (N296) in the last transmembrane helix. These sequence differences above might affect the properties of the enzyme rhodopsins. The schematic model of expressed YFP::RhoPDEs is described in Figure 1, with red asterisks representing the key mutations.

To characterize activities of these RhoPDEs in *Xenopus* oocytes, we fused a YFP tag to the N-terminus of different RhoPDEs, similar to how we characterized *Sr*RhoPDE before, which showed a better expression than the C-terminal YFP fusion construct [15]. After expression of all the YFP::RhoPDEs separately in the same batch of *Xenopus* oocytes, we could observe expressions of most RhoPDEs by the YFP fluorescence, except for *As*RhoPDE, which lacked the key retinal-binding lysine residue (Figure 2). This suggested that lacking the retinal-binding lysine residue in the rhodopsin domain of *As*RhoPDE might influence the protein stability and lead to degradation.

### 3.2. The Hydrolysis Activities of RhoPDEs

After expression of all the eight RhoPDEs in oocytes, the oocyte membrane fragments were extracted to measure the fluorescence emission values and study the enzymatic activities. Fluorescence emissions could be detected for most RhoPDEs except YFP::*As*RhoPDE (Figure 3A). Hydrolysis of cGMP was observed for *Cf*RhoPDE1, *Cf*RhoPDE4 and *Mr*RhoPDE but not in other RhoPDEs.

In our previous study, the *Sr*RhoPDE showed a higher light/dark activity (L/D) ratio of ~4 at a lower initial substrate concentration of ~7.5 μM [15]. Thus, we used ~10 μM cGMP substrate for measuring the activity of new RhoPDEs. Of these, *Cf*RhoPDE1 showed the highest enzyme activity and L/D ratio (~3), whereas *Cf*RhoPDE4 and *Mr*RhoPDE yielded low activity and light-activation (Figure 3B–D).

### 3.3. Light Increased the Substrate Affinity of CfRhoPDE1

We only observed that the L/D activity ratio of YFP::*Cf*RhoPDE1 was increasing from 1.8 to 3 when the initial cGMP concentrations were adjusted from 100 μM to 10 μM. Therefore, we chose to use a range of substrate cGMP from 1 to 100 μM to detect the activity of *Cf*RhoPDE1. The Michaelis–Menten equation was applied to fit the plotted data shown in Figure 4. The maximal hydrolysis velocity was increasing from 580 pmol/min in the dark to 960 pmol/min in the light. However, the Km value for cGMP decreased in the light to ~5.5 μM, from ~15 μM in the dark (Figure 4).

### 3.4. Molecular Engineering of Non-Functional RhoPDEs

When comparing with *Sr*RhoPDE, we identified four of the new RhoPDEs to exhibit differences in key residues, which may lead to the loss of function. Thus, we generated mutations aiming to recover their function. In the rhodopsin domain, *As*RhoPDE lacked the key lysine residue for covalent retinal binding, which may have led to the lacking expression. Therefore, we generated the *As*RhoPDE N296K mutant and then detected a fluorescence when expressed in oocytes (Figure 5).

In the cyclase domain, *Cf*RhoPDE2 exhibited a redundant sequence of 28 amino acids compared with other cyclase domains. We observed that the YFP::*Cf*RhoPDE2 wild type had no cGMP hydrolysis activity. Thus, we deleted this part and then detected cGMP hydrolysis activity of the *Cf*RhoPDE2 deletion mutant, although it had no light regulation (Figure 5). In the *Cf*RhoPDE3 and *Cp*RhoPDE1 catalytic domain, the predicted Mg^2+^-binding site was lacking when compared with functional *Sr*RhoPDE and *Cf*RhoPDE1. Thus, we generated *Cf*RhoPDE3 and *Cp*RhoPDE1 mutants, replacing LRNVG to CHDCD in the catalytic domain, but both mutants showed no cGMP hydrolysis activity. Notably, the four non-functional RhoPDEs (including *Cf*RhoPDE2, 3 and *Cp*RhoPDE1, 2) exhibited shorter N termini (Appendix A), which might participate in the PDE activities and light regulation.

### 3.5. PDE Mutants with Decreased Dark Activity and Unchanged L/D Rratio

After characterization of the eight RhoPDEs, we concluded that *Sr*RhoPDE was still the one with the best light to dark activity ratio (L/D = 4.2). Based on the alignment, we generated another two single mutations of *Sr*RhoPDE close to the substrate binding pocket in the catalytic domain, exhibiting lower cGMP hydrolysis activity in the dark (Figure 6). The YFP::*Sr*RhoPDE L623F mutant activity was decreased to ~1/40 of wt in the dark with a L/D ratio of 3.6, while the E657Q mutant dark activity was reduced to ~1/30 of wt with a L/D ratio of 4.9. Due to the high dark activity of *Sr*RhoPDE, the two mutants might be more appreciated in certain cells where the basal PDE activity is between the dark activity and light activity of the engineered RhoPDEs.

## 4. Discussion

The second messengers cGMP and cAMP play essential roles in many physiological processes. Light-induced nucleotide cyclases were discovered in the last several years and have been well established as optogenetic tools. For cAMP manipulation, photoactivated adenylyl cyclase (PAC) was firstly identified in *Euglena gracilis* and applied in *Drosophila melanogaster* [23] and *Caenorhabditis elegans* [24,25]. Later, bPAC from soil bacterium *Beggiatoa* was characterized with a higher light/dark activity ratio [26]. Recently, bPAC was further engineered to show strongly reduced dark activity, and a new biPAC with reduced dark activity was characterized [27,28]. *Be*Cyclop from *Blastocladiella emersonii* is a green light-activated guanylyl cyclase opsin with a very high L/D ratio [10], ideal for cGMP manipulation. Further guanylyl cyclases are the two-component cyclase opsins (2c-Cyclops) from green algae, either light-inhibited or engineered as light-switchable cyclase opsins [8,12].

However, the current light-regulated PDEs need development, as strong light-regulation of PDE activity is still missing [29]. To this purpose, light-activated phosphodiesterase (LAPD) was first engineered by combining a bacterial photosensor module with the catalytic domain of human PDE2A [30,31]. Upon red-light illumination, LAPD exhibits up to a four-fold and six-fold catalytic activity increase towards cAMP and cGMP, respectively. Preliminary applications showed that LAPD allowed optical control of cAMP and cGMP levels in CHO cells and zebrafish embryos [31]. *Sr*RhoPDE was the first natural photoreceptor discovered with light-enhanced phosphodiesterase activity [14,15]. Current available light-activated PDEs, artificial LAPD and natural RhoPDEs, showed similar drawbacks of relatively high dark activity and low light to dark activity ratio [29]. Therefore, searching and engineering of new light-gated RhoPDEs will provide more possibilities for optogenetic cGMP/cAMP manipulation.

Our previous study showed that fusion of YFP to the N-terminus of *Sr*RhoPDE, provided at least two times stronger fluorescence than the C-terminally fused construct [15]. Therefore, we fused YFP to the N-terminus of each RhoPDE construct and observed expression of most proteins except YFP::*As*RhoPDE. Chop 2 (=Channelopsin-2) K257R lack of retinal binding indeed leads to degradation and complete abolition of light-gated currents [32], probably due to the misfolding when lacking retinal. Similarly, lacking the key retinal-binding lysine residue in the rhodopsin domain might lead to misfolding and degradation of YFP::*As*RhoPDE. The YFP::*As*RhoPDE N296K mutant showed expression, but still no cGMP hydrolysis activity was detected for this mutant. For some other RhoPDEs, differences of key residues can be observed in the PDE catalytic domain, which might interrupt the substrate hydrolysis activity. Then, we generated mutants of these non-functional RhoPDEs and found that *Cf*RhoPDE2 del mutant can recover the cGMP hydrolysis activity but without light regulations. Among the new RhoPDEs tested, *Cf*RhoPDE1 exhibited the highest L/D ratio of ~3.5, and light stimulation could increase the substrate affinity of *Cf*RhoPDE1 from Km = 15 μM to cGMP in the dark to Km = 5.5 μM under illumination.

Sugiura et al. reported that *Mr*RhoPDE showed both cAMP (light-dependent) and cGMP hydrolysis activity after expression in HEK293T cells [19], where the cGMP hydrolysis activity is about four times higher than its cAMP hydrolysis activity. However, we found that the cGMP hydrolysis activity of *Mr*RhoPDE, *Cf*RhoPDE1 and *Cf*RhoPDE4 was over 100 times higher than its cAMP hydrolysis activity after expression in *Xenopus* oocytes. A similar difference can also be seen between our previous studies with *Sr*RhoPDE (SrRh-PDE), where Yoshida et al. detected about a 10 times higher cGMP hydrolysis activity than the cAMP hydrolysis activity for *Sr*RhoPDE (*Sr*Rh-PDE) after expression in HEK293T cells [14], but we saw a difference of about 200 times after expression in *Xenopus* oocytes [15]. The reason underlying this difference is still unclear to us, but we speculate that this could be caused by protein status in different membranes, reaction conditions and detection methods.

Sugiura et al. indicated that *As*Rh-PDE (=*As*RhoPDE) is a cAMP-specific PDE [19]; however, we observed no expression of *As*RhoPDE without the retinal-binding lysine residue. We could recover the expression of *As*RhoPDE in *Xenopus* oocytes with the N296K mutation but cannot recover its cGMP hydrolysis ability. However, we could observe a weak cAMP hydrolysis ability of *As*RhoPDE N296K, which could hydrolyze cAMP from 7.6 µM to 0.5 µM in 90 min and confirmed the cAMP specificity discovered by Sugiura et al., showing that *Cf*Rh-PDE4 (=*Cf*RhoPDE4) exhibited in-cell PDE activity towards cAMP, but no clear activity was observed by HPLC analysis [19]. Here, we observed it is a cGMP specific PDE by immunoassay kit when expressing in *Xenopus* oocytes. A *Xenopus* oocyte is a convenient system for expression of heterologous proteins, but it has also differences to other cells. Ideally, a more accurate conclusion can be made if choanoflagellate cells are used for the function interpretation.

The strong dark activity of *Sr*RhoPDE could pose experimental problems for optogenetic application, as it might lead to a reduction of the cGMP level during the expression in many cell types. Therefore, engineering RhoPDEs with a lower dark activity enables us to fine-tune cGMP/cAMP-regulated pathways. We further generated mutations in the C-terminal catalytic domain and observed that cGMP hydrolysis activities of two single mutants L623F and E657Q decreased over 30 times with unchanged L/D ratios. This could make them applicable in certain cells where the endogenous PDE activity is between its dark activity and light activity.

An open question is how to further improve the efficiency of light regulation. It was reported that the N-terminus of the guanylyl cyclase *Be*Cyclop plays a key role in the light regulation efficiency and enzyme activity [10]. From the structure analysis of *Sr*RhoPDE, the eight transmembrane helices were confirmed. The first transmembrane helix is important for the enzymatic photoactivity [16]. Most non-functional RhoPDEs, such as *Cf*RhoPDE2, 3 and *Cp*RhoPDE1, 2, were predicted with much shorter N-terminal sequences. This might influence their PDE activities and L/D ratios. Further improving the N-termini of rhodopsins may provide the possibility for more promising RhoPDEs as optogenetic tools. From our previous data with *Sr*RhoPDE, the truncated linker regions, between the Rhodopsin domain and PDE domain, also influenced its enzyme activity (unpublished data). Genetically manipulating RhoPDE mutations in the linker regions between the rhodopsin domain and PDE catalytic domain may also provide possibilities to enhance the light regulation efficiency.

## Figures and Tables

**Figure 1 biomolecules-12-00088-f001:**
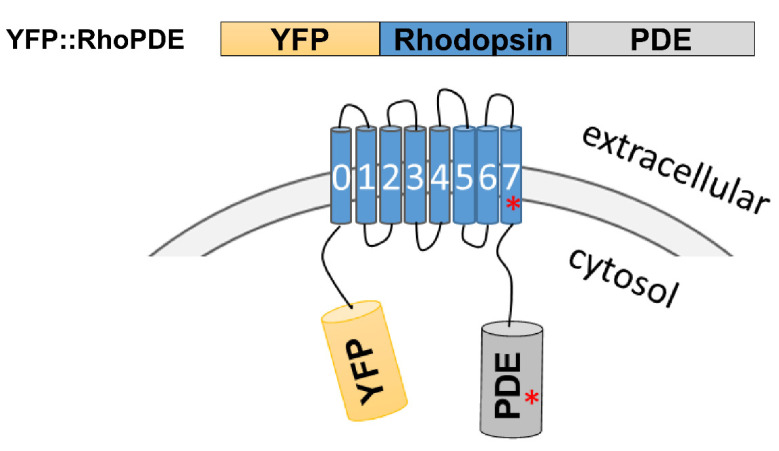
Schematic model of YFP::RhoPDE. The rhodopsin and phosphodiesterase domains were annotated by the PFAM database. *Cf*RhoPDE2, 3 and *Cp*RhoPDE1, 2 were predicted with very short N-terminal sequences. The red asterisks represent some key mutations in the last transmembrane helix or PDE catalytic domain of *As*RhoPDE, *Cf*RhoPDE2, *Cf*RhoPDE3 and *Cp*RhoPDE1.

**Figure 2 biomolecules-12-00088-f002:**
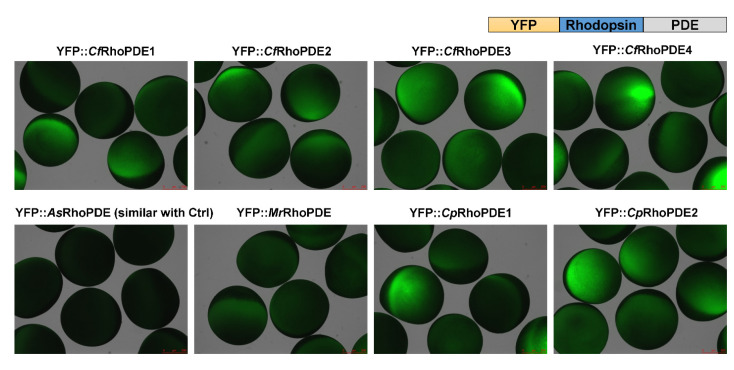
Expression of eight YFP-RhoPDEs in oocytes. For each construct, 30 ng cRNA was injected into oocytes from the same batch. Fluorescence pictures were taken 3 dpi (days post incubation).

**Figure 3 biomolecules-12-00088-f003:**
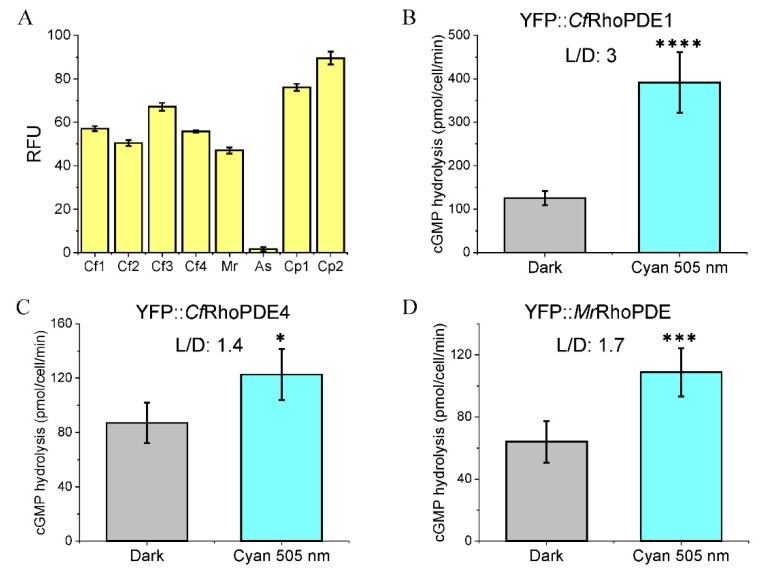
cGMP hydrolysis abilities of YFP::*Cf*RhoPDE1, YFP::*Cf*RhoPDE4 and YFP::*Mr*RhoPDE. (**A**). The relative fluorescence unit (RFU) was measured from membranes of 3 oocytes for each construct. (**B**). cGMP hydrolysis activities of YFP::*Cf*RhoPDE1. (**C**). cGMP hydrolysis activities of YFP::*Cf*RhoPDE4. (**D**). cGMP hydrolysis activities of YFP::*Mr*RhoPDE. For (**B**–**D**), all activities were calculated to membrane proteins extracted from one oocyte. The in vitro reactions were done with 10 μM cGMP. *n* = 4-6, error bars = SD. The two-tailed *t*-test was used in statistical analyses. * *p* < 0.05, *** *p* < 0.001, **** *p* < 0.0001. For each construct, 30 ng cRNA was injected, 3 dpi. The illumination intensity was adjusted to 0.15 mW/mm^2^ with LED light at 505 nm.

**Figure 4 biomolecules-12-00088-f004:**
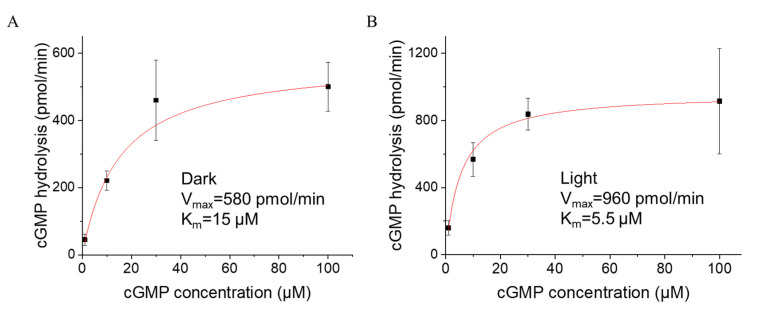
Light altered the substrate binding affinity of YFP::*Cf*RhoPDE1. (**A**), cGMP-dependence of YFP::*Cf*RhoPDE1 activity in the dark. (**B**), cGMP-dependence of YFP:: *Cf*RhoPDE1 activity under illumination with 505 nm cyan light at 0.15 mW/mm^2^. For (**A**,**B**), the data were fitted with the Michaelis–Menten equation. In vitro reactions were started with 1, 10, 30 and 100 μM cGMP. The final activities were calculated for membrane proteins extracted from one oocyte. *n* = 4–6, error bars = SD.

**Figure 5 biomolecules-12-00088-f005:**
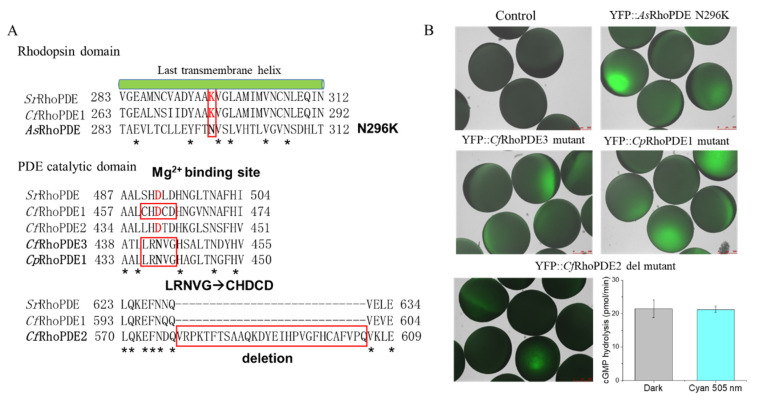
Comparison of key residues of non-functional RhoPDEs. (**A**) Alignment of the non-functional RhoPDEs to functional RhoPDEs in crucial domains. YFP::*As*RhoPDE N296K mutant was generated to reconstitute the retinal-binding lysine in the last transmembrane helix of the rhodopsin domain. In PDE catalytic domains, five residues LRNVG of YFP::*Cf*RhoPDE3 and YFP::*Cp*RhoPDE1 were mutated to a conserved CHDCD motif of functional YFP::*Cf*RhoPDE1 since the D was predicted to be important for the Mg^2+^ binding. (**B**) Fluorescence pictures of mutations from (**A**) and cGMP hydrolysis ability of YFP::*Cf*RhoPDE2 deletion mutant. The final activities were calculated for membrane proteins extracted from one oocyte. The in vitro reactions were performed with 1 μM cGMP. The YFP::*Cf*RhoPDE2 deletion mutant was injected with 10 ng cRNA, 3 dpi. *n* = 6, error bars = SD. The intensity of 505 nm light was adjusted to 0.15 mW/mm^2^.

**Figure 6 biomolecules-12-00088-f006:**
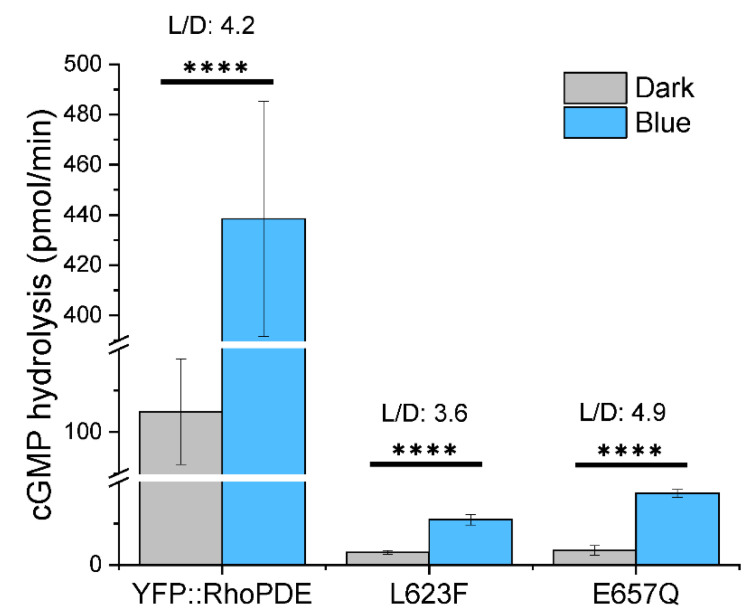
*Sr*RhoPDE L623F and E657Q mutants showed dramatically decreased cGMP hydrolysis activity with similar L/D ratios to wt. In vitro reactions were performed with 10 μM cGMP in the dark and 473 nm blue light illumination at 0.1 mW/mm^2^. Activities of three constructs were adjusted to the same protein amount. *n* = 4, error bars = SD. The two-tailed t-test was used in statistical analyses. **** *p* < 0.0001. The broken *Y*-axis symbol // was introduced from 20 to 90 and from 120 to 390, respectively.

**Table 1 biomolecules-12-00088-t001:** List of primer sequences in 5′-3′ orientation.

Mutations	Forward	Reverse
YFP::*Cf*RhoPDE2 del (28 AA-deletion)	GACCAAGTGAAGCTGG AAAAGCAGCT	GCTTCACTTGGTCGTTGAA CTCTTTCT
YFP::*Cf*RhoPDE3 mutant (441-LRNVG-445 to 441-CHDCD-445)	GATTGTGACCACTCTGCA CTGACCAAC	GTCACAATCGTGACACA GAGTAGCTAACATC
YFP::*As*RhoPDE N296K	CACCAAAGTGTCACTG GTGCACACA	CACTTTGGTGAAGTATTCCA GCAGGCA
YFP::*Cp*RhoPDE1 mutant (436-LRNVG-440 to 436-CHDCD-440)	GATTGTGACCACGCTGGCCT GACCAAC	GTCACAATCGTGACACAGAGCGG CCAGAGCAAG
YFP::*Sr*RhoPDE L623F	CCGTTTCCAAAAAGAGTTC AACAACCAG	CTCTTTTTGGAAACGGGCG CTCCAA
YFP::*Sr*RhoPDE E657Q	AAGGGCCAGATCGGCTTC ATTAACT	GCCGATCTGGCCCTTG GCCAGA

**Table 2 biomolecules-12-00088-t002:** Comparison of different RhoPDEs from choanoflagellates.

RhoPDE Homologs	Identity	Similarity	cGMP/cAMP Hydrolysis	Light Regulation	Sequence Information
*Sr*RhoPDE	100%	100%	Both, cGMP specific	Yes, L/D: 2–4 for both cGMP and cAMP [15]	as the reference sequence for other RhoPDE homologs
*Cf*RhoPDE1	63%	81%	Both, cGMP specific	Yes, L/D: 3 for cGMP	
*Cf*RhoPDE2	37%	57%	No detectable activity	No	additional sequences in the PDE domain in comparison to others
*Cf*RhoPDE3	34%	55%	No detectable activity	No	conserved amino acid in the PDE domain is missing
*Cf*RhoPDE4	46%	64%	Both, cGMP specific	Yes, L/D: 1.4 for cGMP	
*Mr*RhoPDE	79%	91%	Both, cGMP specific	Yes, L/D: 1.7 for cGMP	
*AsRhoPDE*	29%	49%	No detectable activity	No, (no expression)	missing the conserved lysine for ATR binding
*Cp*RhoPDE1	35%	56%	No detectable activity	No	conserved amino acid in the PDE domain is missing
*Cp*RhoPDE2	51%	71%	No detectable activity	No	

The functional RhoPDEs (including *Sr*RhoPDE, *Cf*RhoPDE1, *Cf*RhoPDE4 and *Mr*RhoPDE) weredetermined, while the other RhoPDEs were tested without cGMP hydrolysis activity. ATR: all-trans-retinal. The reference sequence *Sr*RhoPDE NCBI ID: XP_004998010.1. The identity is the number of residues that match exactly between two different sequences. The similarity is the likeness between two sequences with similar properties. The L/D activity ratio was tested under initial 10 μM cGMP.

## Data Availability

The data supporting this research are available upon request.

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
