# Peer review of "Characterization and Modification of Light-Sensitive Phosphodiesterases from Choanoflagellates"

_biomolecules, 2022, doi:10.3390/biom12010088_

Round 1

Reviewer 1 Report

Comments to the authors:

The manuscript entitled “Characterization and modification of light-sensitive Phosphodiesterases from choanoflagellates” reported the results of comprehensive investigations on the cGMP hydrolysis activities of rhodopsin-phosphodiesterases (RhoPDEs) with a view to applying them to optogenetics. The manuscript was easy to understand for readers including non-specialist of this research field. However, there are several concerns to be improved. The authors should make them clear.

Major concerns:

  1. The authors should make sure that all the RhoPDEs and mutant proteins you examined in this study were successfully expressed as functional forms in oocytes. You cannot confirm this only by YFP fluorescence. Using the membrane fraction sample, you can measure visible absorption spectra before and after adding hydroxylamine and make difference spectra. This will enable you to judge whether these proteins were successfully expressed as functional forms or not. Details were described in, for example, Yoshizumi et al. (2016) Biochemistry 55, 5790-5797. See Figure 2 and corresponding Materials and Methods.

If the authors confirm the above, you can improve Table 1, Line 223 “although no light-sensitivity”, Line 226 – 227 “both mutants showed no cGMP hydrolysis activity”, and the relationship between the activity and the short N-terminal sequences in the 4 RhoPDEs that did not show any activity (Line 227 – 228, Line 291-292).

  1. The authors used 505 nm cyan LED light for activation. Is this appropriate for all the RhoPDEs used in this study ? This will also be confirmed after checking the difference absorption spectra described above. If some RhoPDEs absorb blue (~ 480 nm) or green (~ 540 nm) lights, it is inappropriate to interpret the difference in activity based only on the results at 505 nm (Line 180 – 182, Figure 3B-D).

  1. I understand that the enzymerhodopsins with high L/D ratio is recognized as a highly light-regulated ones and useful for optogenetics. However, I cannot understand straightforward the merit that the dark activity is decreased but the L/D ratio is not changed (Line 234 - 236). Isn’t it just a loss-of-function ? In addition, I had an impression that the difference of the L/D ratio 4.2 for wild-type SrRhoPDE and 4.9 for E657Q mutant was not necessarily significant (Figure 6). The authors should address such an objection in Discussion section giving some specific examples (Line 285 - 287).

  1. The authors should make more discussions about experimental data. You can examine the data more closely. For example, why CfRhoPDE-2 & -3 and CpRhoPDE-1 & -2 did not show any activity ?, what is the biological significance of AsRhoPDE ?, what does the change in the Km values for CfRhoPDE-1 in the dark and light states mean ?, what is the role of the short N-terminal sequences ?, what is the difference or similarity between the best studied RhoPDEs and those studied in this study, etc..

Minor concerns:

  1. Provide detailed sequential information, for example sequential alignment, as a supporting information. This will enable readers to easily understand the characteristics, for example the lengths of N-terminal sequences (Line 291 - 292), of RhoPDEs examined in this study.
  2. Figure 2. Fonts for the names of sample shown above each picture were difficult to read.
  3. Figure 5B. For CfRhoPDE2 del. mutant, it is not enough to compare the results of dark and light states. The comparison with the wild-type CfRhoPDE2 is necessary.

Reviewer 2 Report

In this study the authors generated YFP-tagged versions of 8 previously discovered RhoPDEs and assessed their expression in xenopus oocytes. Then they assessed cGMP hydrolysis activity of the constructs with successful expression in the xenopus system (3 out of 7). The authors also employed genetic manipulation approach in an attempt to make some constructs gain cGMP hydrolysis activity (AsRhoPDE, CfPDE2/3 andCpPDE1) and improve the kinetics of SrRhoPDE.  

Major concerns

  1. a) The results are missing comparison in relation to important relevant controls.

- Figure 2 and Figure 3A; why the data was not compared to SrRhoPDE since it has been used as the reference rhodopsin?

- Figure 3B-D: why the cGMP hydrolysis activity of the YFP-tagged RhoPDEs were not compared to the non-YFP tagged counterparts to asses if YFP-fusion alters the level of expression or their light-regulated activity  

- Figure 5B: why the RFU of the mutant versions were not quantified as in Figure 3A and presented in comparison to the corresponding WT version, the same applied to the cGMP hydrolysis activity plot for CfRhoPDE2, why it was not presented in comparison to the WT.

  1. b) There is no mention of the application of any statistical analysis, therefore how did the authors determine if their findings are significant or not?

  1. c) Scientific rational for the experimental design is not provided in some parts

- why enzyme hydrolysis activity for the chosen constructs was determined only for cGMP but not cAMP?

- why enzyme kinetics were only assessed for CfRhoPDE1 and not CfRhoPDE4 or Mr RhoPDE?

- The authors report lack of cGMP hydrolysis activity in WT and mutant versions of CfRhoPDE3, CpRhoPDE1 and AsRhoPDE, this is not surprising as these RhoPEDs have been demonstrated in the original study (https://dx.doi.org/10.1021/acsomega.0c01113 ) to have cAMP specify

- Why did the authors choose to mutate L623 and E657 residues specifically? Do they have a certain/known role in the enzyme catalytic activity, if yes cite reference(s)

  1. d) conclusions and discussion

- The authors suggest that AsRhoPED observed lack of expression could be due to the absence of retinal-binding lysine residue that might influence protein stability.

It seems unlikely that the failure to express the RhoPED is due to the lack of this retinal binding lysine, as expression of functional AsRhoPED has been previously demonstrated (https://dx.doi.org/10.1021/acsomega.0c01113, Figure 3). Moreover, have the  authors tested their hypothesis by applying protein degradation inhibitors?

Also, have the authors considered an alternative explanation for AsRhoPDE failure to produce a signal, for example the fact that it is  chromophore deficient and consequently lacks the ability to absorb/omit light ?  

  • In Table 1, the authors describe sequencing alignment analysis results, however they have not commented if their finding match or differ from what has been reported by Sugiura et al (https://dx.doi.org/10.1021/acsomega.0c01113)
  • The main findings were not discussed in relation to the original study Sugiura et al 2020 or similar studies using the same RhoPDE if they do exist.
  • Figure 2: why the signal is not localized to the rim/membrane of the imaged oocytes since the study is investigating a transmembrane protein?
  • The oocyte images look too prefect and smooth, could the authors provide images of YFP merged with BF
  • What are the key limitations of the study?

Minor comments

1) line 58-59: description of RhoPDE species referred to in this sentence could not be found in the cited transcriptome study [ref 18]  

2) Line 75-76: the source of sequences is not indicated (i.e. NCBI transcript ID/ref seq ID), alternatively if these RhoPDs were obtained from de novo sequencing please indicate how?

3) It would be helpful to mention the other names for these RhoPDEs as per ref [19] the original study

4) The exact position of the altered residues (point mutations/deletions) is missing from the methods section

5) the details about the site directed mutagenesis kit and the primers sequences used to introduce the changes are not supplied

6) how SrRhoPDE mutants were generated is not mentioned in the methods

7) line 120-121: did the authors conduct imaging on live or fixed cells?

8) Table 1:

-What is the difference between similarity and identity in sequence alignment? including a definition in the table foot note will be helpful

- how could the rhodopsin be cGMP specific/ and non-specific at the same time? (i.e. if it is capable of breaking down both types of second messenger)

 - The data/results upon which the information on cAMP/cGMP light regulation of SrRhoPDE (L/D ratio) is not supplied in the manuscript

- how the L/D was calculated and what does it signify in terms of suitability and application of the RhoPDEs

- Definition of ATR binding is not included.

- what does No mean? no expression or property not tested?

-references from which the rhodopsin structural domains information was obtained are not included

9) The L/D values reported in Figure 3 are not consisted with Table 1.

10) lines 227-228: the authors report that the four non-functional RhoPDEs exhibited shorter N termini, it would be more informative if a figure showing sequence alignment of the non-functional constructs against SrRhoPDE was added to demonstrate the differences between the length of the N-termini

11) Figure 6: the y-axis scaling does not allow the reader to infer the values of each column because it uses 10x multiplication as units, instead the authors can redo the figure and introduce a break to accommodate the higher values.

12) line 268: cNMP definition is not provided

13) lines 284-286: would not reducing the dark (spontaneous activity) have a technical advantage i.e better signal to noise ratio?

14) lines 293-294: “From our previous data with SrRhoPDE, the truncated linker regions, between the Rhodopsin domain and PDE domain, also influenced its enzyme activity.” Citation of the  data source is missing

15) lines 295-297: the closing statement is not clear. Did the authors mean screening for naturally occurring mutants of the RhoPDE(s) or genetically manipulating these enzymes in the lab will shed light on how to improve their performance?

Round 2

Reviewer 1 Report

Thank you to all the authors who responded to my comments. The manuscript has been improved to some extent, but there is still room for improvement in the experimental design and the way the results are presented and discussed. I hope that this will be useful for your future research.

Reviewer 2 Report

 Author's Reply to the Review Report (Reviewer 2) 

 Comments and Suggestions for Authors 

 In this study the authors generated YFP-tagged versions of 8 previously discovered RhoPDEs and assessed their expression in xenopus oocytes. Then they assessed cGMP hydrolysis activity of the constructs with successful expression in the xenopus system (3 out of 7). The authors also employed genetic manipulation approach in an attempt to make some constructs gain cGMP hydrolysis activity (AsRhoPDE, CfPDE2/3 andCpPDE1) and improve the kinetics of SrRhoPDE.   

 Major concerns 

 a) The results are missing comparison in relation to important relevant controls. 

 Figure 2 and Figure 3A; why the data was not compared to SrRhoPDE since it has been used as the reference rhodopsin?  

Answer: Thanks for your comments. Previously, we already characterized SrRhoPDE with the same expression system and in vitro reaction condition in the cited reference [15]. So here we mainly compared the other eight RhoPDEs instead of SrRhoPDE. We think it is not necessary to repeat previous test with the same methods in this work. 

Reviewer reply:  Since the authors already have the data for SrRhoPDE, can they plot it against the rest of the constructs in Figure 3A and insert the images for SrRhoPDE for the sake of comparison in Figure 2

Figure 3B-D: why the cGMP hydrolysis activity of the YFP-tagged RhoPDEs were not compared to the non-YFP tagged counterparts to asses if YFP-fusion alters the level of expression or their light-regulated activity   

Answer: Based on the homolog SrRhoPDE, we already compared the cGMP hydrolysis activity with or without YFP tag in the reference [15]. We observed N-terminal fused YFP tag showed much better expression than the C-terminal YFP fusion construct. Moreover, the non-YFP tagged SrRhoPDE showed the similar light regulation effect with YFP::SrRhoPDE. From this experience, here we screen all homologous constructs of RhoPDEs only with N-terminal fused YFP tag. This would be useful to compare the activity of different mutants in the similar expression level.  

Reviewer reply: We thank the authors for the clarification  

Figure 5B: why the RFU of the mutant versions were not quantified as in Figure 3A and presented in comparison to the corresponding WT version, the same applied to the cGMP hydrolysis activity plot for CfRhoPDE2, why it was not presented in comparison to the WT. 

Answer: For the RFU tests, we already subtracted the background value from WT version (no cRNA injection) without any fluorescence. We observed that YFP::CfRhoPDE2 WT has no the cGMP hydrolysis activity, so the null values are not plotted. 

Reviewer reply:  Oocytes that have not been injected with cRNA (no cRNA injection) and those injected with WT versions of the constructs are two different conditions. no cRNA are oocytes that have not been injected with anything serve as negative controls. In order to assess the effect of any mutant version of a construct it has to be in relation to its WT version (i.e. the oocyte injected with the WT “un-manipulated” version). According to Figure 3A the WT versions of CfRhoPDE3/2 and CpRhoPDE1 generate delectable signals, therefore they could be compared to their mutant counterparts. As for CfRhoPDE2 cGMP hydrolysis activity comparison with WT version, the provided clarification should be included in the main text (i.e. the authors observation that the WT version lacks cGMP hydrolysis activity)

 There is no mention of the application of any statistical analysis, therefore how did the authors determine if their findings are significant or not? 

Answer: Thanks for your comments. We supplemented the two-tailed T-test for the statistical analysis in the updated Figure 3 and Figure 6 and figure legends. So our findings about light activation of RhoPDEs are significant.  

Reviewer reply:  We thank the authors for the clarification  

Scientific rational for the experimental design is not provided in some parts 
 why enzyme hydrolysis activity for the chosen constructs was determined only for cGMP but not cAMP? 

Answer: We tested the cAMP hydrolysis activities of all the constructs and only found that YFP::CfRhoPDE1, 4 and YFP::MrRhoPDE have the quite low enzymatic activity. For cGMP in vitro reaction, it takes few minutes to hydrolyze cGMP from 10 μM to below 1 μM. But for cAMP in vitro reaction, we observed 1-2 μM cAMP was hydrolyzed in 60 min. Furthermore, we cannot observe any light regulation in the cAMP hydrolysis assay. Therefore, we only choose to determine cGMP hydrolysis instead of cAMP. 

Reviewer reply:  
The authors are advised to include this explanation where appropriate in the text for the benefit of the reader and to provide rational for their study design
The authors are encouraged to discuss their findings in relation to Sugiura et al study (https://dx.doi.org/10.1021/acsomega.0c01113), as they have shown MrRhoPDE to have hydrolysis activity towards both cAMP (light-dependent) and cGMP, also they have shown AsRhoPDE to have cAMP-specific activity (Figure 3). Can the authors speculate the reasons for this discrepancy?

 why enzyme kinetics were only assessed for CfRhoPDE1 and not CfRhoPDE4 or Mr RhoPDE? 

 Answer: In this work, we used different initial cGMP concentrations (100 μM, 10 μM) to test the three functional constructs YFP::CfRhoPDE1, 4 and YFP::MrRhoPDE. We only observed the L/D activity ratio of YFP::CfRhoPDE1 was increasing from 1.8 to 3, when the initial cGMP concentrations were adjusted from 100 μM to 10 μM. But YFP::CfRhoPDE4 and YFP::MrRhoPDE have no L/D ratio change under different cGMP concentrations. Therefore, we only assessed for the enzyme kinetic of YFP::CfRhoPDE1. 

 Reviewer reply:  

 The authors are advised to include this explanation where appropriate in the text for the benefit of the reader and to provide rational for their study design
Also the authors are encouraged to discuss their findings in relation to Sugiura et al study (https://dx.doi.org/10.1021/acsomega.0c01113), as they have shown MrRhoPDE to have what seems like light-independent cGMP  hydrolysis activity which is similar to what is observed here. However, CfRhoPDE4 seems to have light-regulated cAMP and cGMP hydrolysis activity (Sugiura et al  Figure 3) which contradicts with what has been observed here.  Can the authors include a comment on this point where relevant in the main text. 

 The authors report lack of cGMP hydrolysis activity in WT and mutant versions of CfRhoPDE3, CpRhoPDE1 and AsRhoPDE, this is not surprising as these RhoPEDs have been demonstrated in the original study (https://dx.doi.org/10.1021/acsomega.0c01113) to have cAMP specify 

Answer: Similar with CfRhoPDE2, we tested WT and mutant versions of CfRhoPDE3, CpRhoPDE1 and AsRhoPDE, but no cGMP hydrolysis can be detected. So we did not put the null values for figures. We found that YFP::CfRhoPDE1, 4 and YFP::MrRhoPDE have the quite low enzymatic activity without light regulation effect. So we only choose to determine cGMP hydrolysis activity which is over 100 times than cAMP hydrolysis activity from our tests. 

Reviewer reply:  

The authors are encouraged to discuss their findings in relation to the original study in which the constructs were first described (https://dx.doi.org/10.1021/acsomega.0c01113)

Why did the authors choose to mutate L623 and E657 residues specifically? Do they have a certain/known role in the enzyme catalytic activity, if yes cite reference(s) 

Answer: We choose to mutate the PDE catalytic domain in order to decrease the dark activity of SrRhoPDE. We only generated the two mutations based on structure of homologous PDE catalytic domain close to the substrate binding pocket. There is no relative reference but picked by ourselves according to the alignment. 

Reviewer reply:  The authors are advised to include this explanation where appropriate in the text for the benefit of the reader and to provide rational for their study design

d) conclusions and discussion 

- The authors suggest that AsRhoPED observed lack of expression could be due to the absence of retinal-binding lysine residue that might influence protein stability. 

It seems unlikely that the failure to express the RhoPED is due to the lack of this retinal binding lysine,   as  expression       of         functional AsRhoPED            has       been     previously       demonstrated 

(https://dx.doi.org/10.1021/acsomega.0c01113, Figure 3). Moreover, have the authors tested their hypothesis by applying protein degradation inhibitors? 

Answer: We think the main difference is the protein yield of the two protein expression systems. Previously AsRhoPDE was demonstrated with cAMP hydrolysis activity when expressing in HEK293 cells. Here we use a different expression system Xenopus oocyte, and the protein yield is quite low, if any. Different host system often leads to varied expression level, due to different molecule machinery in hosting cells. In addition, lack of retinal binding indeed could lead to instability of many microbial rhodopsins, as stated in the previous publications of our lab, Ullrich, Sybille et al. “Degradation of channelopsin-2 in the absence of retinal and degradation resistance in certain mutants.” Biological chemistry vol. 394,2 (2013): 271-80. We expect that the AsRhoPDE N296K mutant should also show higher expression level in HEK 293 cells. We have tried protein degradation inhibitors like MG-132 in Xenopus oocytes before but did not found obvious improvement. 

Reviewer reply: 
The authors are advised to include this explanation where appropriate in the text and to cite the previous study (Ullrich et al 2013)
Also can the authors speculate the degradation mechanism(s) through which the retinal binding lysine residue provides resistance against degradation since MG-132 had no effect?  

 Also, have the authors considered an alternative explanation for AsRhoPDE failure to produce a signal, for example the fact that it is chromophore deficient and consequently lacks the ability to absorb/omit light?   

 Answer: We detect the protein amount through the YFP tag. The chromophore of YFP is buried inside of the β-barrels, it’s unlikely that AsRhoPDE would lead to chromophore deficiency of YFP. Therefore, we suspect that lacking YFP signal should be a result of protein instability of AsRhoPDE, at least in Xenopus oocytes system. 

 Reviewer reply: We thank the authors for the clarification  

 In Table 1, the authors describe sequencing alignment analysis results, however they have not commented if their finding match or differ from what has been reported by Sugiura et al (https://dx.doi.org/10.1021/acsomega.0c01113) 

 The main findings were not discussed in relation to the original study Sugiura et al 2020 or similar studies using the same RhoPDE if they do exist. 

 Answer: (Note: In methods section, we added the list of primer sequences as the Table 1. So the original Table 1 was replaced by Table 2 in the results section.) We added the alignment sequences in figure S1. The sequences we used match the report from Sugiura et al. We further focus on the comparison of YFP-tagged constructs and corresponding mutations. 

 We used the Xenopus oocytes as the expression system while Sugiura et al. used the HEK293 cells. So it is reasonable to observe some different results due to different physiological conditions. Sugiura et al. did not detect the activity of CfRhoPDE4, but we can observe the cGMP hydrolysis of YFP::CfRhoPDE4. AsRhoPDE can be expressed in HEK293 cells but not the case in oocytes. The fluorescence was recovered in the YFP::AsRhoPDE N296K mutant in oocytes. But similar results showed CfRhoPDE1, CfRhoPDE4, and MrRhoPDE exhibit light-dependent PDE activity.  

 Reviewer reply:

 “Sugiura et al. did not detect the activity of CfRhoPDE4, but we can observe the cGMP hydrolysis of YFP::CfRhoPDE4”                                                                                          Sugiura et al have detected a cGMP  hydrolysis activity for  CfRhoPDE4 but it may not be very obvious as their plot scale is high (Figure 3)
“But similar results showed CfRhoPDE1, CfRhoPDE4, and MrRhoPDE exhibit light-dependent PDE activity”                                                                                                                      This statement is not clear. In Siguira et al, MrRhoPDE shows light-dependent activity towards cAMP, and light-insensitive activity towards cGMP, but in the presented study the authors report that it lacks cAMP hydrolysis activity and report light-dependent activity towards cGMP?. Moreover, Siguira et al demonstrated that CfRhoPDE1 and CfRhoPDE4 exhibited detectable light-dependent activity towards cAMP, which is not in agreement with what is reported here. Can the authors provide explanation and include it in the discussion?

 Figure 2: why the signal is not localized to the rim/membrane of the imaged oocytes since the study is investigating a transmembrane protein? 

 Reviewer reply: We thank the authors for the clarification  

 The oocyte images look too prefect and smooth, could the authors provide images of YFP merged with BF 

 Reviewer reply: We thank the authors for the clarification  

 What are the key limitations of the study? 

 Reviewer reply: The authors have not discussed the study key limitations in the revised version  

 Answer: In this study, we checked the fluorescence by using the normal fluorescence microscope instead of confocal one because of the availability. The Xenopus oocytes are relatively big and not transparent, even the soluble protein looks ring-like under confocal microscope. Furthermore, we think it more convincing to extract the membrane and measure the fluorescence of the membrane extract. Then main fluorescence measurement was detected from extracted membrane fraction via Fluoroskan Ascent microplate fluorometer. 

 We prepared the fluorescence images which is already merged with BF in Figure 2, here take images of YFP::CfRhoPDE1 as an example. YFP::CfRhoPDE1 images of BF, Fluo, merged are shown below: 

 Reviewer reply: We thank the authors for the clarification  

 Minor comments 

 line 58-59: description of RhoPDE species referred to in this sentence could not be found in the cited transcriptome study [ref 18]   

Answer: These species can be found in Figure 3A consensus phylogenetic tree of [ref 18]. 

Reviewer reply: We thank the authors for the clarification  

Line 75-76: the source of sequences is not indicated (i.e. NCBI transcript ID/ref seq ID), alternatively if these RhoPDs were obtained from de novo sequencing please indicate how? 

Answer: We thank Dr. N. King of UC Berkeley providing the sequences information of RhoPDEs. At the moment, some RhoPDE sequences cannot be found in NCBI.  

Reviewer reply: Can the authors include this statement in the methods section and comment if the sequences can be provided to interested readers upon request?

It would be helpful to mention the other names for these RhoPDEs as per ref [19] the original study 

Answer: Thanks for your suggestions. We put the other names as ref [19] in the introduction, e.g. CfRhoPDE1=CfRh-PDE1 

Reviewer reply: We thank the authors for the clarification  

The exact position of the altered residues (point mutations/deletions) is missing from the methods section 

Answer: Thanks for your comments. We put the exact position of the altered residues in the method section of the revised manuscript. 

 Reviewer reply: We thank the authors for the clarification  

the details about the site directed mutagenesis kit and the primers sequences used to introduce the changes are not supplied 

Answer: We used the quickchange mutagenesis method. Here we added a Table 1 with primers sequences to introduce those mutations. 

 Reviewer reply:  Please add the exact kit name as there are different types and versions

how SrRhoPDE mutants were generated is not mentioned in the methods 

Answer: We still used the quickchange mutagenesis method and added the primer sequences in Table 1. 

Reviewer reply:  Kindly indicate that where relevant in the methods section 

line 120-121: did the authors conduct imaging on live or fixed cells? 

Answer: we conducted imaging on live cells. 

 Reviewer reply:  Kindly indicate that where relevant in the methods section 

Table 1:  

-What is the difference between similarity and identity in sequence alignment? including a definition in the table foot note will be helpful 

Answer: (Note: In methods section, we added the list of primer sequences as the Table 1. So the original Table 1 was replaced by Table 2 in the results section.) 

We put a definition of similarity and identity in the table foot note. After protein sequences alignment, the similarity is the likeness between two sequences, and similar sequences contain similar properties. The identity is the number of residues that match exactly between two different sequences. 

Reviewer reply: We thank the authors for the clarification  

 how could the rhodopsin be cGMP specific/ and non-specific at the same time? (i.e. if it is capable of breaking down both types of second messenger) 

Answer: We tested that YFP::CfRhoPDE1, 4 and YFP::MrRhoPDE have the quite low cAMP hydrolysis activity without light regulation effect. We determined cGMP hydrolysis activity over 100 times than cAMP hydrolysis activity. So the functional RhoPDEs show cGMP specific and cAMP non-specific hydrolysis activity. 

Reviewer reply: The authors are encouraged to include this as a clarification of how they defined enzyme substrate specificity in their study

The data/results upon which the information on cAMP/cGMP light regulation of SrRhoPDE (L/D ratio) is not supplied in the manuscript 

Answer: We added the light regulation of SrRhoPDE (L/D ratio) and put the ref [15] in the updated Table 2. 

Reviewer reply: We thank the authors for the clarification  

how the L/D was calculated and what does it signify in terms of suitability and application of the 

RhoPDEs 

Answer: After extracting certain number of oocytes membrane fractions, we did in vitro reaction mixed with reaction buffer including certain concentrations of cGMP or cAMP. Under dark or light conditions, the decreasing concentrations of cGMP/cAMP was measured by Chemiluminescent Immunoassay Kit in certain time points. So the light and dark activity and L/D activity ratio can be calculated. The ideal RhoPDE would be the high L/D ratio with lower dark activity. This would make it quite useful in the physiological roles of different cell types to regulate cyclic nucleotides.  

Reviewer reply
 It would be very helpful to include this information in the methods section
It is not clear what the authors are trying to say in the last statement “This would make it quite useful in the physiological roles of different cell types to regulate cyclic nucleotides”

 Definition of ATR binding is not included. 

Answer: We define the ATR as all-trans retinal in the foot note.  

Reviewer reply: We thank the authors for the clarification  

what does No mean? no expression or property not tested? 

Answer: No means we cannot detect the cGMP/cAMP hydrolysis activity. We changed “No” to “No detectable activity” in the updated Table 2. The light regulation is still put “No” because no light regulation can be detected. 

 Reviewer reply: We thank the authors for the clarification  

-references from which the rhodopsin structural domains information was obtained are not included Answer: In the footnote, we added the reference sequence SrRhoPDE with NCBI ID XP_004998010.1. It is easy to obtain the sequence information from database. 

Reviewer reply: We thank the authors for the clarification  

 The L/D values reported in Figure 3 are not consisted with Table 1. 

Answer: We give some explanation of L/D in the foot note of updated Table 2. Different initial substrate concentrations have effects on the L/D ratio of CfRhoPDE1. In Figure 3, we only used initial 10 μM cGMP for in vitro reactions, and the L/D ratio is ~3. We used initial 1 μM cGMP with L/D ~3.5, but initial 100 μM cGMP was decreasing L/D ratio at ~1.8. These data are used in Figure 4 under different substrate concentrations.  

Reviewer reply: We thank the authors for the clarification  

lines 227-228: the authors report that the four non-functional RhoPDEs exhibited shorter N termini, it would be more informative if a figure showing sequence alignment of the non-functional constructs against SrRhoPDE was added to demonstrate the differences between the length of the N-termini 

Answer: Thanks for your good suggestions. We prepared the supplemental figure S1 to align the rhodopsin sequences among all the RhoPDEs to figure out that the N-terminal variation could be one of elements to regulate enzymatic activities. 

 Reviewer reply: We thank the authors for the clarification  

Figure 6: the y-axis scaling does not allow the reader to infer the values of each column because it uses 10x multiplication as units, instead the authors can redo the figure and introduce a break to accommodate the higher values. 

Answer: Thanks for your good suggestions. We changed the figure 6 to make it easy to read the values. 

Reviewer reply: We thank the authors for the clarification  

line 268: cNMP definition is not provided Answer: we changed as you suggested. 

 Reviewer reply: We thank the authors for the clarification  

lines 284-286: would not reducing the dark (spontaneous activity) have a technical advantage i.e better signal to noise ratio? 

Answer: Basically, if we want the dynamic manipulation, then the lower dark activity is necessary. Otherwise the background activity is too high in the dark already. If the dark activity is below the basal endogenous activity and cannot influence the physiology condition of certain cell types, then the light illumination time can be used to fine-tune the cGMP/cAMP regulated signaling pathways, thus controlling the cell behaviors or investigating other physiological roles. 

Reviewer reply: Do the authors mean that having a lower dark activity allows for easier manipulation of enzyme activity and subsequently better control of cGMP/cAMP regulated pathways? if yes then it would benefit the reader to include this in the discussion

lines 293-294: “From our previous data with SrRhoPDE, the truncated linker regions, between the Rhodopsin domain and PDE domain, also influenced its enzyme activity.” Citation of the data source is missing 

Answer: This is our pervious data unpublished. We truncated linker regions of SrRhoPDE and found out the different enzyme activity but no light regulation. So these mutations cannot be used as optogenetic tools.  

Reviewer reply: Can the authors indicate in the text that this statement is based on observations from unpublished data

lines 295-297: the closing statement is not clear. Did the authors mean screening for naturally occurring mutants of the RhoPDE(s) or genetically manipulating these enzymes in the lab will shed light on how to improve their performance? 

Answer: We changed the statement clear. We would say that genetically manipulating these enzymes in the lab could improve their performance.  

Reviewer reply: We thank the authors for the clarification  
